# Mechanism of the Wake-Up and the Split-Up in AlO*_x_*/Hf_0.5_Zr_0.5_O*_x_* Film

**DOI:** 10.3390/nano13142146

**Published:** 2023-07-24

**Authors:** Min-Jin Kim, Cheol-Jun Kim, Bo-Soo Kang

**Affiliations:** 1Department of Applied Physics, Hanyang University, Ansan 15588, Republic of Korea; kim.minzzin@gmail.com; 2Department of Applied Physics, Center for Bionano Intelligence Education and Research, Hanyang University, Ansan 15588, Republic of Korea; chjun0926@hanyang.ac.kr

**Keywords:** ferroelectric, hafnium oxide, first order reversal curves, energy landscape, switching mechanism, split up, landau theory, MIFM capacitor

## Abstract

Dielectric layers are widely used in ferroelectric applications such as memory and negative capacitance devices. The wake-up and the split-up phenomena in the ferroelectric hafnia are well-known challenges in early-stage device reliability. We found that the phenomena even occur in the bilayer, which is composed of the hafnia and the dielectrics. The phenomena are known to be affected mainly by oxygen vacancies of hafnia. Dielectric layers, which are often metal oxides, are also prone to be affected by oxygen vacancies. To study the effect of the dielectric layer on the wake-up and the split-up phenomena, we fabricated ferroelectric thin-film capacitors with dielectric layers of various thicknesses and measured their field-cycling behaviors. We found that the movement of oxygen vacancies in the dielectric layer was predominantly affected by the polarization state of the ferroelectric layer. In addition, the mechanism of the field-cycling behavior in the bilayer is similar to that in ferroelectric thin films. Our results can be applied in ferroelectric applications that use dielectric layers.

## 1. Introduction

Interestingly, some dielectrics have spontaneous polarization even when there is no electric field applied, and the direction of polarization can be reversed by the polarity of the external field. These are called ferroelectrics. The most primary characteristic of ferroelectrics is the polarization-voltage (*P*–*V*) hysteresis loop. Upon applying an ac voltage, the polarization exhibits a non-linear hysteretic curve depending on the external voltage amplitude. As-grown ferroelectric material starts at P=0, where the polarization is randomly oriented, and the total polarization inside the film is zero. Once a sufficient voltage is applied and polarization is aligned, the ferroelectric has the polarization value on the *P*–*V* hysteresis loop as the applied voltage is varied. The axis intercepts are the key parameters in this hysteresis loop, that is, the remnant polarization $P_{r}$ and coercive voltage $V_{c}$. A voltage larger than $V_{c}$ is needed in order to reverse the polarity of the polarization. The value of $P_{r}$ is a criterion by which ferroelectrics can be used in non-volatile memories. However, conventional perovskite-based ferroelectrics often lose their ferroelectricity when scaled down, which is called the finite size effect.

Ferroelectric HfO_2_ thin film has been investigated with various fabrications for applications because of their scalability and compatibility with the traditional complementary metal-oxide-semiconductor [1,2,3,4,5,6,7,8,9,10,11,12,13]. Particularly for reliability in applications such as memory devices and negative capacitance devices, the evolution of the *P*–*V* hysteresis loop has been studied during polarization switching cycles [14,15,16,17]. The ferroelectric properties discovered in HfO_2_ thin film have solved the integration and scalability issues in the applications, but there are still some obstacles in device reliability. The large coercive field (1MV/cm) of HfO_2_ associated with a large free energy barrier between bi-stable polarization states reduces the retention loss for the non-volatile memory device. But, it needs a high field to switch the polarization, which would cause the evolution of the *P*–*V* loop.

Oxygen vacancies play a critical role in their evolution since they move or are generated under switching cycles. These induce wake-up and fatigue phenomena, indicating the increment and the decrement in the $P_{r}$, respectively [14,16]. Studying the split-up effect, which causes the pinching of the *P*–*V* loop, is helpful for understanding the movement of oxygen vacancies. Schenk et al. reported that oxygen vacancies are affected by the electric field induced by the polarization state in the static domain during a subcycle, and that the split-up effect can be induced by various local domain free energy landscapes [15,16]. The oxygen vacancies inside the domain affect the free energy landscape of the domain. Various domains with different polarization states and oxygen vacancy distributions contribute to different free energy landscapes. If the landscapes have two distinct tendencies, the split-up effect occurs when the entire thin film is switched.

Figure 1 shows a scenario in which the internal bias field $E_{bias}$ is zero because all the domains have a uniform distribution of oxygen vacancies. The polarization of the ferroelectric domain switches from negative to positive when the applied electric field across it is greater than the switching field $E_{s}$, and vice versa when the applied field is less than the backswitching field $E_{bs}$ [16]. When the amplitude of the applied field cycling $\left|E\right|$ is greater than $\left|E_{bs,1}\right|$ but less than $\left|E_{s,1}\right|$, which are the absolute values of the backswitching and switching fields for Domain 1, Domain 1 has negative polarization, and the oxygen vacancies move toward the bottom electrode owing to the electric field induced by the polarization state in Domain 1. The oxygen vacancies at the interface induce a positive $E_{bias}$, which tilts the free energy landscape of Domain 1 to the left. Similarly, the landscape of Domain 2 is tilted to the right. The electric field induced by these static domains can also affect the oxygen vacancies in neighboring domains [16]. Consequently, the split-up effect is induced by the free energy landscapes with various values of $E_{bias}$.

The insertion of a dielectric layer in a ferroelectric-based device is important for applications [3,18,19,20,21,22,23,24]. The purpose of the insertion can be diverse. First, the dielectric/ferroelectric bilayer can be used as a main operation part in ferroelectric-based memory devices [22]. In this case, the insertion of the dielectric layer improved the leakage and endurance properties in memory applications. Second, the dielectric layer also can serve as a tunneling barrier in ferroelectric tunnel junctions [18,23]. Incorporating an extra dielectric layer forms structural asymmetry, resulting in the different potential barrier height and tunneling resistances. Third, the negative capacitance can be expected [3,19,20,21,24]. The intervened dielectric layer at one near-electrode interface would induce both a depolarization field across the ferroelectric layer and an internal field across the dielectric layer. Therefore, dielectric/ferroelectric bilayers should be studied rigorously in a practical point of view.

Compared to a single ferroelectric layer, the movement of oxygen vacancies might be more complicated in a dielectric/ferroelectric bilayer because of the additional interface, the dielectric layer polarization, and the oxygen–atom exchange between the two layers. Therefore, the movement of oxygen vacancies in the bilayer should be discussed in order to understand the mechanism of the field-cycling behavior in the bilayer.

The free energy landscape of the bilayer can be expressed as the sum of the free energy functions of the dielectrics and the ferroelectrics [19]. The local free energy of the bilayer may be affected by the distribution of oxygen vacancies. Kim et al. proposed that the local free energies can be obtained from the switching density distribution [25]. We therefore measured the distributions of the switching densities and investigated the movement mechanism of the oxygen vacancies in the bilayer in the pristine, wake-up, and split-up states.

In this study, we fabricated a metal-ferroelectric-metal (MFM) capacitor, which has a standard ferroelectric capacitor structure and metal-insulator-ferroelectric-metal (MIFM) capacitors. We investigated the evolutions of the *P*–*V* hysteresis and the switching current−voltage (*I*–*V*) curves and changes in the switching densities of the samples under polarization switching cycles. The local free energy landscapes were obtained from the distributions of the switching densities, and the movement mechanism of the oxygen vacancies is discussed. The similarity between the mechanisms of the MFM and MIFM capacitors indicates that the mechanism of the field-cycling behavior can also be applied to the ferroelectric-dielectric bilayer. Our results explain the mechanism of the field-cycling behavior in MIFM capacitors and are also applicable to ferroelectric-based applications with dielectric layers.

## 2. Materials and Methods

Ten-nanometer-thick hafnium zirconium oxide (HZO) films were grown on a TiN-coated Si substrate, and subsequently, the aluminum oxide (AO) layers with thicknesses varying from 0 to 3 nm were deposited using atomic layer deposition. The fabricated devices consisted of the structures as Figure 2. The AO was deposited in situ after HZO was deposited to prevent the external impurities. The deposition temperature for the AO was 300 °C in that the HZO was deposited due to the good thermal stability of the trimethylaluminum we used. TiN top electrodes were deposited by sputtering using a shadow mask. To achieve the ferroelectricity of the HZO, all the samples were annealed at 600 °C for 1 min using a rapid thermal process.

The *P*–*V* loops and *I*–*V* curves were measured at 1 kHz with a bipolar triangular waveform using a Precision LC Ⅱ ferroelectric tester (Radiant Technologies, Inc., Albuquerque, NM, USA). Small-signal capacitance−voltage (*C*–*V*) measurements were performed at an amplitude of 50 mV using an E4980A LCR meter (Agilent Technologies, Inc., Santa Clara, CA, USA).

To evoke the wake-up effect, all samples in the pristine state were subjected to the voltage cycles. The voltage cycles were provided by a rectangular wave with a frequency of 1 kHz for 1 s. The voltage height was varied according to the thickness of the dielectric layer. It is known that the transient current reaches its maximum value around the coercive field and gradually declines at the field that is lower than the coercive field. At the field where the transient current finally vanishes, all domains are completely switched. We name this field the saturated switching field. To achieve the split-up effect, the wave with amplitude in the range between the coercive voltage and the saturated switching voltage was applied across the samples in the wake-up state [26]. For the appropriate analysis of the split-up behavior, the amplitude of the subcycles is defined based on the position of the switching current peak in the *I*–*V* curve obtained from the sample in the wake-up state.

By fitting the polarization $P\left(V,V_{r}\right)$ data measured using the ferroelectric tester using the first-order reversal curves method, the switching density $\rho \left(V,V_{r}\right)$ was obtained as [16]
(1)$$\rho \left(V,V_{r}\right)=-\frac{1}{2}\frac{\partial^{2}P\left(V,V_{r}\right)}{\partial V\partial V_{r}},$$
where V is applied voltage and $V_{r}$ is the reversal voltage. To calculate $\rho \left(V,V_{r}\right)$, we extract the consecutive $P\left(V,V_{r}\right)$ data point on the $\left(V,V_{r}\right)$ plane. These points are fitted with a second-order polynomial surface of the form $a_{1}+a_{2}V+a_{3}V^{2}+a_{4}V_{r}+a_{5}V_{r}^{2}+a_{6}VV_{r}$; then, $-0.5a_{6}$ is taken as the value of $\rho \left(V,V_{r}\right)$. The number of data points on a local grid is ${\left(2SF+1\right)}^{2}$, where SF is the smoothing factor. Here, SF is set as 1, which corresponds to a 3×3 grid [27].

The polarization curves were obtained from numerous V sweeps from $V_{r}$ to the maximal voltage $V_{max}$, where $V_{r}$ was varied from $V_{max}$ to $-V_{max}$. $\rho \left(V,V_{r}\right)$ was converted into $\rho \left(V_{bias},V_{c}\right)$ using $V_{bias}$ and $V_{c}$ [16]:(2)$$V_{bias}=\frac{V+V_{r}}{2},V_{c}=\frac{V-V_{r}}{2},$$
where $V_{bias}$ is the internal bias voltage. In $\left(V_{bias},V_{c}\right)$ configuration, it is easy to identify the switching behaviors as the results of cycling. Also, the notations used here—internal bias voltage and coercive voltage—are more familiar since those are also common parameters usually discussed in *P*–*V* loops.

In the Landau–Ginzburg–Devonshire (LGD) theory, the free energy per volume of a ferroelectric u is given as a polynomial function of P [28]:(3)$$u=\alpha P^{2}+\beta P^{4}+E_{bias}P.$$
where α and β are the ferroelectric anisotropy constants. According to the LGD theory, the term $–E_{eff}P$ consisting of the product of the effective field and the polarization describes the electrostatic energy. The effective field is expressed in terms of the external applied field and the internal bias field as $E_{eff}=E_{app}-E_{bias}$. Here, no external field is applied across the film to observe without external signals. Thus, the effective field in the film is $-E_{bias}$ [28]. Therefore, only $E_{bias}$ remains in the linear term of P. The values of α and β can be obtained from the remnant polarization and coercive field [25]:(4)$$\alpha =-\frac{3\sqrt{3}E_{c}}{4P_{r}},\beta =\frac{3\sqrt{3}E_{c}}{8P_{r}^{3}}.$$

As mentioned earlier, the free energy landscape for an MIFM capacitor can be expressed as the sum of the landscapes for the dielectrics and ferroelectrics [19]. Therefore, $E_{bias,MIFM}$ and $E_{c,MIFM}$ can be extracted from the landscape for the MIFM capacitor. However, they represent neither the properties of the ferroelectric layer nor those of the dielectric layer. Here, $E_{bias,MIFM}$ and $E_{c,MIFM}$ are simply the internal bias voltage and the coercive voltage divided by the total thickness of the bilayer, respectively.

## 3. Results

### 3.1. Pristine, Wake-Up and Split-Up

Figure 3 shows the *P*–*V* loops and *I*–*V* curves of all the samples. Owing to the voltage division, higher voltages were necessary to switch the polarization as the AO layer became thicker. When the amplitude of the triangular wave was 3 V, a decrease of $2P_{r}$ was observed with a thicker AO layer (this is not shown here). In particular, $2P_{r}$ of 3 nm-AO/HZO capacitor is about $4\mu C{cm}^{-2}$. In the memory application, the reasonable magnitude of $2P_{r}$ is to be larger than $6\mu C{cm}^{-2}$ [16]. In such a context, it may seem like 3 nm-AO/HZO thin film is improper for use in memory devices. But, a larger operation voltage would enhance the $2P_{r}$ value. To facilitate the proper observation of the ferroelectric properties in each capacitor, the maximum values of the applied voltage were adjusted to 2.5 V, 3 V, 4 V and 5.0 V for 0 nm-, 1 nm, 2 nm and 3 nm AO/HZO samples, respectively.

Despite the insertion of the dielectric layer, all the samples exhibited the wake-up and split-up effects. In the wake-up effect, the *P*–*V* loop is widened in the vertical direction, and the peak of the *I*–*V* loop is increased. In the split-up effect, the *P*–*V* loop is pinched, and the original peak of the *I*–*V* loop splits into two peaks. It indicates that a polarization switching similar to what happened in the MFM capacitors took place also in the MIFM capacitors.

Due to the low field cycling, only the domains that have the switching field lower than the cycling field were switched. When the current was measured using a higher amplitude field, the switching of the remaining domains that were not switched during the subcycles was revealed as a current peak split. Thus, the current peak in the wake-up state split into two peaks, one at the lower field and the other at the higher field.

As shown Figure 4, the MIFM capacitor in the pristine state exhibits the typical ‘butterfly’ *C*–*V* curve observed in MFM capacitors. Because a thin dielectric layer with a large capacitance was connected in series with a ferroelectric layer, most of the voltage was applied to the ferroelectric layer. Therefore, the ferroelectric layer dominated the *C*–*V* curve of the MIFM capacitor. The *C*–*V* curve in the split-up state is pinched in a similar manner to the *P*–*V* loop.

Figure 5 shows the switching density distributions in all the samples. The distribution changes in the MIFM capacitors follow a trend similar to those in the MFM capacitors. In the wake-up state, the maximum switching density increased, while the width of the distribution was maintained. This indicates that more domains participated in switching. In the split-up state, the peak of the switching density split into two peaks with the different $V_{bias}$ values in a similar manner to the *I*–*V* curve. One peak is the more negative $V_{bias}$, and the other is around zero. This is because the samples in the pristine state already had a negative $V_{bias}$, and the bias fields maintained the negative values for the wake-up. In other words, the switching density peak in the wake-up is split into two peaks in opposite directions on the $V_{bias}$-axis.

### 3.2. Free Energy Landscapes

Figure 6 shows the free energy landscapes of the local domains for all the samples. $E_{bias}$ and $E_{c}$ were used when the corresponding switching densities exceeded 60% of the maximum switching density because they were most actively involved in switching. In Figure 5, colors ranging from black to blue denote the different landscapes in the pristine and wake-up states. The landscapes of the domains in these states exhibited similar tendencies and changed gradually. The landscapes in blue and red are based on the bias features in the split-up state.

In the wake-up states, the landscapes are scaled up but still tilted to the right. This indicates that $E_{bias}$ had almost negative values in the switching densities of the pristine and wake-up states. In contrast, some of the energy landscapes in the split-up state are tilted to the left, and others are tilted more to the right, while they scaled down. The involvement of domains belonging to two landscape groups during the polarization switching of the MFM and MIFM capacitors resulted in the split-up effect.

## 4. Discussion

All the MIFM capacitors exhibited ferroelectric characteristics with both the wake-up and split-up effects like in MFM capacitors, despite the presence of dielectric layers. The *P*–*V* loop had the larger $2P_{r}$ due to the wake-up effect and was pinched due to the split-up effect. The peak of the *I*–*V* curve was increased due to the wake-up effect and split into two peaks due to the split-up effect. The distribution of switching densities shows that more domains participated in the polarization switching due to the wake-up effect, and the peak of that split into two peaks due to the split-up effect. The *C*–*V* curve showed typical ferroelectric characteristics because a thin dielectric layer with a large capacitance was connected in series with a ferroelectric layer. The curve is pinched due to a split-up effect in a similar manner to the *P*–*V* loop.

The wake-up effect in HfO_2_ films is induced by phase transitions. The monoclinic and tetragonal phases, which are paraelectric, turn into the ferroelectric orthorhombic phase when the oxygen vacancies are redistributed. The oxygen vacancies diffuse or drift due to alternating field cycling and are uniformly distributed throughout the ferroelectric thin film. The process dominates in the wake-up stage. Later, as the fatigue effect occurs, the generation of new oxygen vacancies dominates the polarization switching properties [14]. The presence of a wake-up effect in the MIFM capacitor therefore indicates that the redistribution of oxygen vacancies in the ferroelectric layer plays a dominant role in the polarization switching cycles.

The switching density peak in the wake-up state is split along opposite directions into the two peaks of the split-up state. As shown in Figure 1, the splitting phenomenon originates from the different internal bias fields induced by oxygen vacancies in the film. The splitting phenomenon thus indicates that oxygen vacancies have moved to either interface. If the oxygen vacancies were trapped at the interface between the dielectric and ferroelectric layers, the internal bias field of the dielectric layer would be opposite to that of the ferroelectric layer, which is unlikely considering the observed split-up effects. In other words, such vacancies cannot result in a tilted free energy landscape. Therefore, the movement of oxygen vacancies toward the interfaces with the electrodes was dominant during the subcycles. This may be because the local polarization of the ferroelectric layer affected that of the dielectric layer at the corresponding site. The polarization of the local dielectric layer is hard to switch during the subcycles because the corresponding ferroelectric domain has static polarization. The polarization of the dielectrics induces the constant electric field. The oxygen vacancies move depending on the field. If the polarization is positive, the oxygen vacancy moves toward the top electrode/dielectric interface. It makes the landscape tilt to the right. Otherwise, if the polarization is negative, the oxygen vacancy in the dielectric layer moves toward the dielectric/ferroelectric interface. Previously, trapping at the interface between the dielectric and ferroelectric layers was not dominant. Then, since the static domain also has the negative polarization, the oxygen vacancy moves toward the ferroelectric/bottom electrode interface. It makes the landscape tilt to the left. As a result, the static domains in the ferroelectric layer with positive or negative polarizations induce the split-up effect in the MIFM capacitor.

Oxygen vacancy has been considered important in many metal oxides. The activation energy of oxygen vacancy diffusion in alumina is known to be relatively high. However, according to the density functional theory calculation, unlike non-charged oxygen vacancy, the charged vacancy has low activation energy. Furthermore, the energy level of the oxygen vacancy in Al_2_O_3_ matches well with the fermi level of TiN, and the oxygen vacancy in our thin alumina is adjacent to the electrode. Thus, the diffusion of the charged oxygen vacancy at the operation voltage of 1–2 V is well facilitated [29]. This supports our experimental results.

During full cycling, the oxygen vacancies are redistributed uniformly in both layers [14]. During the subcycles, the electric field induced by polarization in the static domain sends oxygen vacancies toward either of the two interfaces. The movement of the oxygen vacancies in the MIFM capacitor is identical to that in the MFM capacitor, in that the oxygen vacancies eventually move to the interfaces with the electrodes. In addition, all the MIFM capacitors exhibited ferroelectric switching characteristics, such as wake-up and split-up effects. Therefore, the mechanism of the field-cycling behavior can be applied to a dielectric/ferroelectric bilayer during switching cycles.

## 5. Conclusions

We fabricated an MFM capacitor and MIFM capacitors in which the thicknesses of the dielectrics were 1, 2 and 3 nm. To investigate the effect of the insertion of the dielectric layer in the ferroelectric capacitor, we obtained the *P*–*V* loops, *I*–*V* curves, *C*–*V* curves, and the distributions of the switching densities. The samples had the significant $2P_{r}$ values and distinct peaks of the switching current and switching density. The MIFM capacitor also exhibited the typical butterfly *C*–*V* curve observed in the MFM capacitor.

Also, high- and low-amplitude field cycles were applied across the samples for the sake of the wake-up and split-up effects. At high-amplitude field cycles, $2P_{r}$ is larger in the *P*–*V* loop, the peak is increased in the *I*–*V* curve, and more domains participated in polarization switching in the distribution of switching densities. We found that the MIFM capacitors exhibited wake-up effects, despite the insertion of the dielectric layer. This indicated that the uniform redistribution of oxygen vacancies in the ferroelectric layer played a dominant role in the MIFM capacitor. Since the polarization switching of the dielectric layer was affected by that of the ferroelectric layer, the oxygen vacancies were redistributed uniformly in the dielectric layer as well.

At low-amplitude field cycles called subcycles, the *P*–*V* loop and *C*–*V* curve were pinched, and the original peaks of the *I*–*V* curve and switching density split into two peaks. We also obtained the free energy landscapes from the switching density. The original landscapes were tilted to the right. At low-amplitude field cycles, we found that some were tilted to the left, and others were tilted more to the right. This is because the oxygen vacancies moved toward the two interfaces with the top and the bottom electrodes. In other words, the static polarization of the dielectric layer induces the electric field, and the oxygen vacancies move to the interface with the electrode under the field.

Our results indicate that oxygen vacancies were uniformly redistributed throughout the ferroelectric–dielectric bilayer at high-amplitude field cycles, while they moved to the interfaces with the electrodes at low-amplitude field cycles. The resultant movement of the oxygen vacancies was identical to that in the MFM capacitor. Therefore, we explained the mechanism of the field-cycling behavior in the bilayer. Our results are applicable to ferroelectric-based applications with dielectric layers.

## Figures and Tables

**Figure 1 nanomaterials-13-02146-f001:**
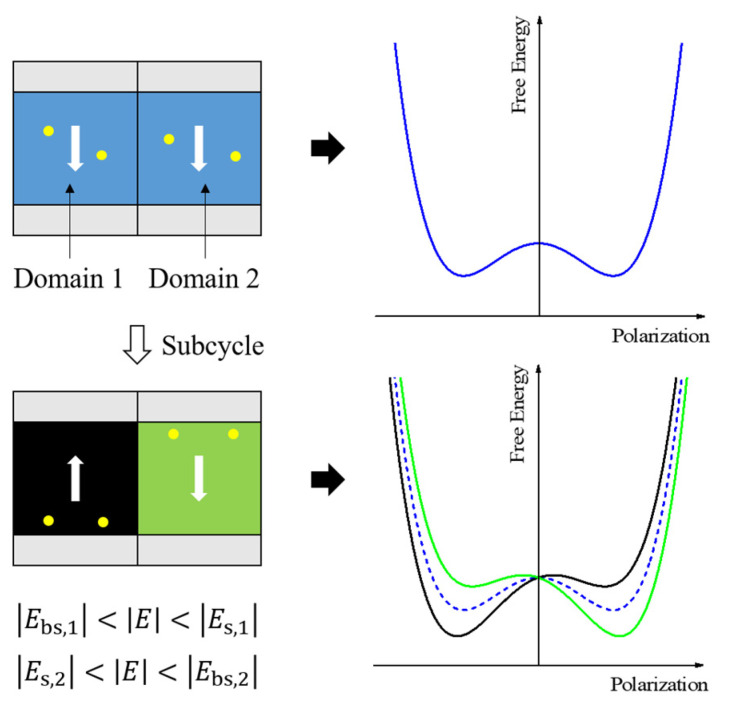
The free energy landscapes at different domain states. The landscape corresponding to each domain is indicated by the same color. The white arrow and the yellow dot in the domain represent the polarization and the oxygen vacancy, respectively. The split-up effect is induced by local landscapes with different $E_{bias}$.

**Figure 2 nanomaterials-13-02146-f002:**
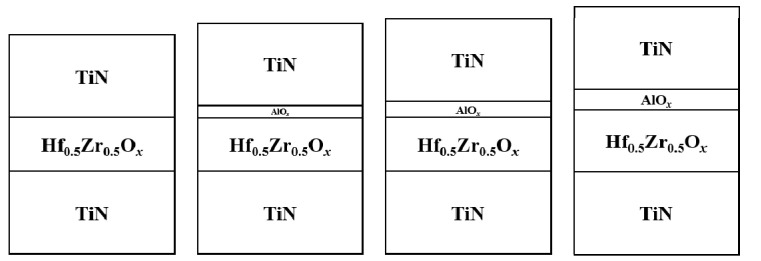
Schematics of the fabricated devices.

**Figure 3 nanomaterials-13-02146-f003:**
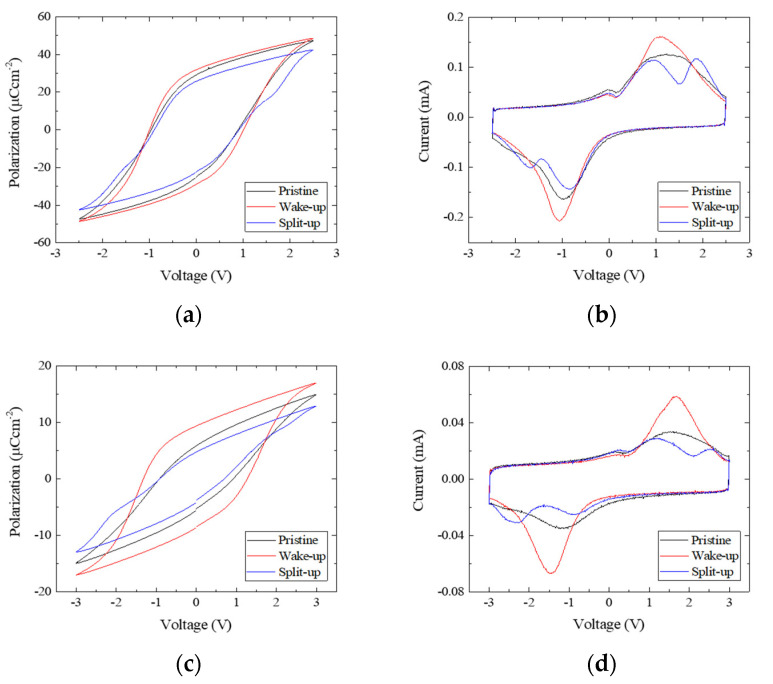
The *P*–*V* and *I*–*V* loops for (**a**,**b**) the MFM capacitor and the MIFM capacitors with AO thicknesses of (**c**,**d**) 1 nm, (**e**,**f**) 2 nm, and (**g**,**h**) 3 nm. In the wake-up, $P_{r}$ and the maximum of the switching current are increased. In the split-up, *P*–*V* loop is pinched and the peak of *I*–*V* loop is split into two peaks.

**Figure 4 nanomaterials-13-02146-f004:**
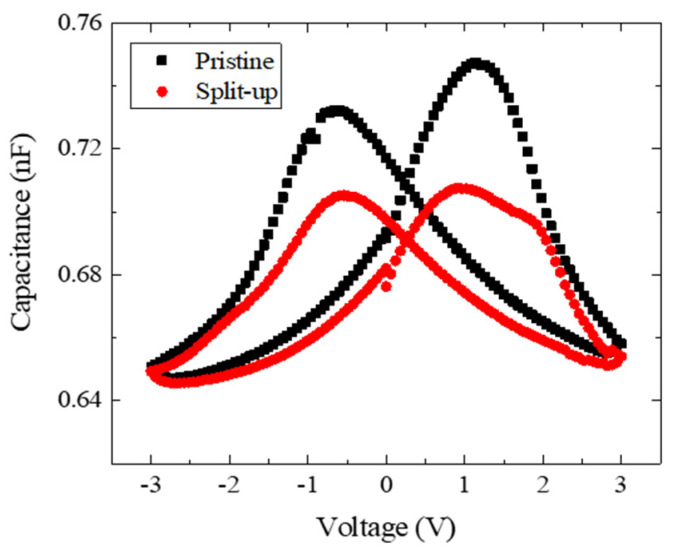
The small-signal *C*–*V* characteristics of the MIFM capacitor with 1 nm AO in the pristine and split-up states.

**Figure 5 nanomaterials-13-02146-f005:**
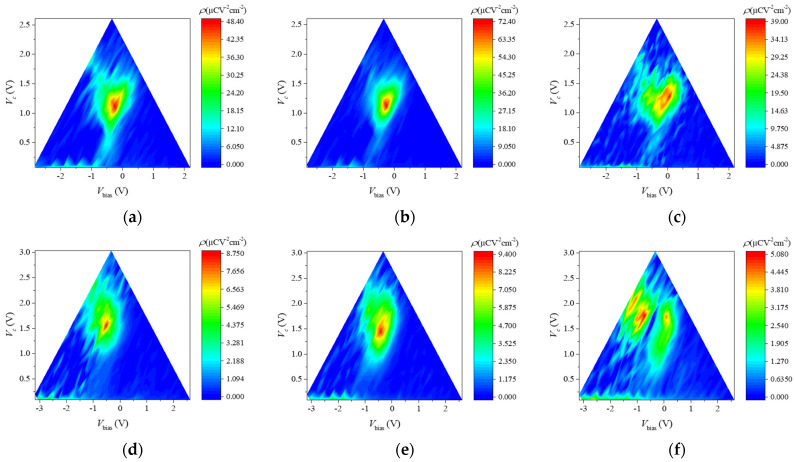
Switching density distributions in the pristine, wake-up and split-up states for (**a**–**c**) the MFM capacitor and for the MIFM capacitors with the AO thickness of (**d**–**f**) 1 nm; (**g**–**i**) 2 nm; and (**j**–**l**) 3 nm, respectively. The peak of the switching density increased in the wake-up and split into two peaks in the split-up.

**Figure 6 nanomaterials-13-02146-f006:**
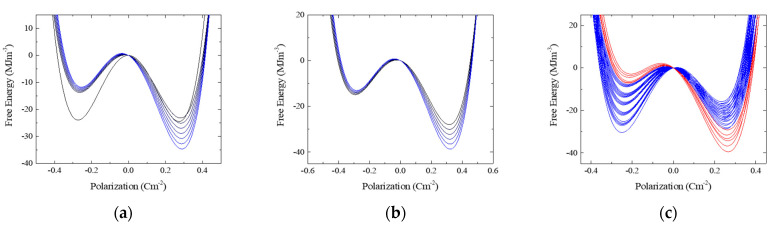
The free energy landscapes in the pristine, wake-up and split-up states for (**a**–**c**) the MFM capacitor and for the MIFM capacitors with the AO thickness of (**d**–**f**) 1 nm; (**g**–**i**) 2 nm; and (**j**–**l**) 3 nm, respectively. In the wake-up, the landscapes are scaled up. In the split-up, the landscapes are divided into two groups according to the bias tendencies of them. The two groups were colored blue and red, respectively. In the pristine and wake-up states, on the other hand, the landscapes with the similar bias tendency were colored ranging from black to blue.

## Data Availability

The data presented in this study are contained within the article.

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
