# Peer review of "Mechanism of the Wake-Up and the Split-Up in AlOx/Hf0.5Zr0.5Ox Film"

_nanomaterials, 2023, doi:10.3390/nano13142146_

Round 1
Reviewer 1 Report
In the present manuscript Kim et al. have studied the effect of including a dielectric aluminum oxide dielectric layer between the ferroelectric hafnium oxide layer to understand the role of oxygen vacancy migration during field cycling. In additional the authors have also analyzed the I-V, C-V and P-V loops from different thicknesses of the inserted aluminum oxide to optimize the layer thickness. The results are interesting to the readership of Nanomaterials, especially to the oxide ferroelectric community. But the authors should address the following points prior to further consideration for publication.
Major
1. Please provide a cross sectional TEM image of the full stack. Also please confirm if the thin Al2O3 is crystalline, else please refer to it as AlOx as the standard non-stoichiometric amorphous phase, which should also be clear from the TEM image.
2. The authors emphasize on the redistribution/migration of the oxygen vacancy towards the electrode interfaces after cycling. The authors should characterize this change via XPS measurements before and after low-amplitude field cycles to see a clear change in the oxygen vacancy at the oxide top/electrode interface, by simply repeating the experiments with a thin top TiN electrode, enabling probing the XPS depth. This would confirm the chemical migration claimed at least during the low-amplitude field cycling for the electrical characterizations.
Minor
3. In the Materials and Methods section, please include the deposition conditions for the Al2O3 layers as this is known to have an effect on the oxide and electrode interfaces
4. Please include a schematic sketch of the fabricated MFIM device
Author Response
Point 1: Please provide a cross sectional TEM image of the full stack. Also please confirm if the thin Al2O3 is crystalline, else please refer to it as AlOx as the standard non-stoichiometric amorphous phase, which should also be clear from the TEM image.
Response 1: It would be best if TEM measurement could relate the oxygen vacancy and the split-up phenomena. But, the oxygen vacancy distribution regarding the split-up should be considered not only in the in-plane TEM image but also in the cross-section image. It may not be easy to focus on important local sites. We thought the first-order reversal curves (FORCs) methods allow us to understand the distribution.
We corrected Al2O3 to AlOx. Considering the oxygen vacancy, we also corrected Hf0.5Zr0.5O2 to Hf0.5Zr0.5Ox.
Point 2: The authors emphasize on the redistribution/migration of the oxygen vacancy towards the electrode interfaces after cycling. The authors should characterize this change via XPS measurements before and after low-amplitude field cycles to see a clear change in the oxygen vacancy at the oxide top/electrode interface, by simply repeating the experiments with a thin top TiN electrode, enabling probing the XPS depth. This would confirm the chemical migration claimed at least during the low-amplitude field cycling for the electrical characterizations.
Response 2: We believe it would be good if we could provide a direct evidence by using XPS. But, we observed the phenomena under the micrometer-scale electrode in the large sample and the X-ray beam size is typically millimeter-scale. The XPS will give the averaged response for the beam spot, and therefore will not show clear change in the oxygen signal at the interface. However, the FORCs methods allow us to get an overall picture.
Point 3: In the Materials and Methods section, please include the deposition conditions for the Al2O3 layers as this is known to have an effect on the oxide and electrode interfaces.
Response 3: Aluminum oxide was deposited in situ after hafnium zirconium oxide (HZO) was deposited to prevent the external impurities. The deposition temperature is 300℃ in that HZO was deposited due to a good thermal stability of trimethylaluminum we used. We added the sentences in the manuscript.
Point 4: Please include a schematic sketch of the fabricated MFIM device.
Response 4: We added the schematics as the Figure 2.
Reviewer 2 Report
My decision is: Accept after minor revision (corrections to minor methodological errors and text editing).
The topic is interesting and the results are well described. Some minor points need attention:
- The authors should reorganize the Title and Abstract to emphatically extrude the novelty of this study. The abstract is not concise and can't reflect the main idea of the work.
- Please refer to the latest progress report as far as possible, and the "Introduction" part should be concise.
- Why the authors did not perform Mott-Schottky measurements for analyzing the oxygen vacancies behavior in Al2O3?
- There are some formatting mistakes in the references section, I suggest the authors check and correct them. For example, there are incomplete references or with erroneous data, others with typos in the journal name, or chemical formulae in the title, e.g. refs# 10, 17, 20, and 22.
Minor editing of English language required.
Author Response
Point 1: The authors should reorganize the Title and Abstract to emphatically extrude the novelty of this study. The abstract is not concise and can't reflect the main idea of the work.
Response 1: We changed the Title to “Mechanism of the wake-up and the split-up in AlOx/Hf0.5Zr0.5Ox film” to extrude emphatically the novelty. Also, we modified the Abstract to reflect the main idea.
Point 2: Please refer to the latest progress report as far as possible, and the "Introduction" part should be concise.
Response 2: The latest references were added and the “Introduction” part was concisely modified.
Point 3: Why the authors did not perform Mott-Schottky measurements for analyzing the oxygen vacancies behavior in Al2O3?
Response 3: We suppose Mott-Schottky measurements would be great if we had one Schottky interface. But, in our sample with top and bottom interfaces, we were not sure about the analysis. We should do the further studies by performing the Mott-Schottky measurements.
Point 4: There are some formatting mistakes in the references section, I suggest the authors check and correct them. For example, there are incomplete references or with erroneous data, others with typos in the journal name, or chemical formulae in the title, e.g. refs# 10, 17, 20, and 22.
Response 4: We checked and corrected all them.
Reviewer 3 Report
This article is devoted to the study of switching processes in ferroelectric layers Hf0.5Zr0.5O2, on which an additional layer of dielectric Al2O3 was grown. A number of important results were obtained in the article, showing the key role of oxygen vacancy diffusion in switching processes. The obtained results ensure the relevance of the article and its scientific significance. For a better understanding of the role of oxygen vacancy diffusion, I have two small comments.
1. The article very briefly states that after annealing (lines 120-125), the resulting films become crystalline. It is necessary to characterize the studied samples in more detail in order to understand the degree of their crystallinity. Are these polycrystalline films or not, or is one film epitaxial and the other polycrystalline? If possible, it is necessary to provide evidence of their crystallinity (XRD, RHEED, Raman or something else). Does the annealing time affect their crystallinity and the results obtained in the article?
2. I assume that the processes of diffusion of oxygen vacancies in such materials have been studied theoretically by the density functional method (DFT). It would be important to briefly describe these results and provide a few references to recent works.
This article is devoted to the study of switching processes in ferroelectric layers Hf0.5Zr0.5O2, on which an additional layer of dielectric Al2O3 was grown. A number of important results were obtained in the article, showing the key role of oxygen vacancy diffusion in switching processes. The obtained results ensure the relevance of the article and its scientific significance. For a better understanding of the role of oxygen vacancy diffusion, I have two small comments.
1. The article very briefly states that after annealing (lines 120-125), the resulting films become crystalline. It is necessary to characterize the studied samples in more detail in order to understand the degree of their crystallinity. Are these polycrystalline films or not, or is one film epitaxial and the other polycrystalline? If possible, it is necessary to provide evidence of their crystallinity (XRD, RHEED, Raman or something else). Does the annealing time affect their crystallinity and the results obtained in the article?
2. I assume that the processes of diffusion of oxygen vacancies in such materials have been studied theoretically by the density functional method (DFT). It would be important to briefly describe these results and provide a few references to recent works.
Author Response
Point 1: The article very briefly states that after annealing (lines 120-125), the resulting films become crystalline. It is necessary to characterize the studied samples in more detail in order to understand the degree of their crystallinity. Are these polycrystalline films or not, or is one film epitaxial and the other polycrystalline? If possible, it is necessary to provide evidence of their crystallinity (XRD, RHEED, Raman or something else). Does the annealing time affect their crystallinity and the results obtained in the article?
Response 1: We annealed the hafnium zirconium oxide (HZO) to get the ferroelectric orthorhombic phase. HZO is known to show ferroelectricity when it is polycrystalline. The ferroelectricity of our HZO is a evidence that HZO is polycrystalline film including the orthorhombic phase. We modified “To achieve the crystalline” to “To achieve the ferroelectricity of HZO” in the manuscript for clarity. Alumium oxide was expected to play the role of a general dielectric layer, so the crystallinity of the aluminum oxide was not analyzed.
Point 2: I assume that the processes of diffusion of oxygen vacancies in such materials have been studied theoretically by the density functional method (DFT). It would be important to briefly describe these results and provide a few references to recent works.
Response 2: Thanks to the reviewer’s comment, we could add a reference supportive to our results. We added the proper reference and the brief decription was inserted into the Discussion in the page 10 of the revised manuscript. The decription is as follows. “Oxygen vacancy has been considered important in many metal oxides. The activation energy of oxygen vacancy diffusion in alumina is known to be relatively high. However, according to the density functional theory calculation, unlike non-charged oxygen vacancy, the charged vacancy have low activation energy. Furthermore, the energy level of the oxygen vacancy in Al2O3 matches well with the fermi level of TiN, and the oxygen vacancy in our thin alumina is adjacent to the electrode. Thus, the diffusion of charged oxygen vacancy at the operation voltage of 1-2 V is well facilitated [APPLIED PHYSICS LETTER 103, 093504 (2013)]. This supports our experimental results.“
Reviewer 4 Report
The paper is devoted for effect of dielectric layer on polarization switching characteristics of Al2O3/Hf0.5Zr0.5O2 film. The topic is generally interesting, however the paper contain unexplained places (below) and need major revisions.
The aim of the paper should be clearly formulated.
Figs. 2, 4 and 5 should be more commented. Why some curves in Fig. 5 are drawn with the same color and some other with different?
Lines 17-18 ‘’We found that the movement of oxygen vacancies in the dielectric layer was predominantly affected by the polarization state..’’ I not find evidence for this in the paper text.
Conclusions should be rewritten in more informative way.
English need minor revisions.
English need minor revisions.
Author Response
Point 1: Figs. 2, 4 and 5 should be more commented. Why some curves in Fig. 5 are drawn with the same color and some other with different?
Response 1: We added the additional figure as Figure. 2, and Figs. 2, 4, and 5 were modified to Figs. 3, 4, and 6. We added the descriptions of the corresponding data in Figs. 3, 4, and 6. In Fig. 6, there are the local free energy landscapes in the MIFM capacitors. The capacitors in the pristine and the wake-up were switched continuously under the applied voltage. We checked up the landscapes of the capacitors and they had similar bias tendencies. Therefore, they were drawn in the same color. On the other hand, in the capacitors in the split-up, the first switching at lower voltage is followed by the second switching at higher voltage. The discrete switching must be originated from the different bias features of the landscapes. We checked up the landscapes and they were drawn in two colors depending on their bias features.
Point 2: Lines 17-18 ‘’We found that the movement of oxygen vacancies in the dielectric layer was predominantly affected by the polarization state..’’ I not find evidence for this in the paper text.
Response 2: The supportive statements are presented over the “Discussion” part. The movement of oxygen vacancies in the ferroelectric (FE) layer according to the FE polarization is well known. But, if the movement ended up at the dielectric(DE)/FE interface, the split-up phenomena wouldn’t be observed. So, the split-up phenomena show that the movement of the oxygen vacancies happens all the way to the DE/electrode interface. This is why the movement of oxygen vacancies in the DE layer is predominantly affected by FE polarization.
Point 3: Conclusions should be rewritten in more informative way.
Response 3: The “Conclusion” part was rewritten in more information way.
Point 4: English need minor revisions.
Response 4: Some awkward sentences were modified.
Round 2
Reviewer 1 Report
I am satisfied with the author's response to my queries and can now recommend the manuscript for publication in Nanomaterials.
Reviewer 4 Report
Authors make proper corrections according to reviewer remarks and I suggest to publish the paper as it is.